# MMLSCU: A Dataset for Multimodal Multi-domain Live Streaming Comment Understanding

## ABSTRACT

With the increasing popularity of live streaming, the interactions from viewers during a live streaming can provide more specific and constructive feedback for both the streamer and platform. In such scenario, the primary and most direct feedback method from the audience is through comments. Thus, mining these live streaming comments to unearth the intentions behind them and, in turn, aiding streamers to enhance their live streaming quality is significant for the well development of live streaming ecosystem. To this end, we introduce the MMLSCU dataset, containing 50,129 intention-annotated comments across multiple modalities (text, images, videos, audio) from eight streaming domains. Using multimodal pretrained large model and drawing inspiration from the Chain of Thoughts (CoT) concept, we implement an end-to-end model to sequentially perform the following tasks: viewer comment intent detection ⇒ intent cause mining ⇒ viewer comment explanation ⇒ streamer policy suggestion. We employ distinct branches for video and audio to process their respective modalities. After obtaining the video and audio representations, we conduct a multimodal fusion with the comment. This integrated data is then fed into the large language model to perform inference across the four tasks following the CoT framework. Experimental results indicate that our model outperforms three multimodal classification baselines on comment intent detection and streamer policy suggestion, and one multimodal generation baselines on intent cause mining and viewer comment explanation. Compared to the models using only text, our multimodal setting yields superior outcomes. Moreover, incorporating CoT allows our model to enhance comment interpretation and more precise suggestions for the streamers. Our proposed dataset and model will bring new research attention on multimodal live streaming comment understanding.

## CCS CONCEPTS

• **Information systems** → **Multimedia information systems**; • **Computing methodologies** → **Natural language generation**; **Language resources**.

## KEYWORDS

Live Streaming, Multimodal, Comment Understanding

**ACM Reference Format:**
. 2018. MMLSCU: A Dataset for Multimodal Multi-domain Live Streaming Comment Understanding. In *Proceedings of Make sure to enter the correct conference title from your rights confirmation emai (Conference acronym' XX).* ACM, New York, NY, USA, 12 pages. https://doi.org/XXX.XXXX

## 1 INTRODUCTION

In the current era, live streaming has emerged as one of the dominant methods for content distribution, drawing a substantial number of streamers and viewers to participate. As depicted in Figure 1, which showcases the live streaming platform Twitch[1], streamers are delivering personalized live content to their viewers. Beyond merely watching, viewers actively post comments to engage in the live streaming sessions, expressing reactions for the live content. Such real-time comments also provide streamers valuable feedback, allowing for dynamic adjustments to content or strategies, thereby establishing a robust interaction loop between streamers and viewers. Given this context, mining the comments of live streaming holds practical value, contributing to both multimodal content understanding and the advancement of the live streaming industry.

However, it is difficult to understand and parse the unique community culture, as it encompasses a large amount of non-standard vocabulary, domain-specific jargon, memes, as well as oral expressions, and a variety of emojis [33]. In response to these challenges, some preliminary research has been conducted. Wang et al. [38] proposed a video comment multimodal dataset without any annotation, and the authors only suggested a comment generation task, lacking in-depth exploration of the content in the comment dataset. Similar issues exist in works such as [5, 26]. Xu et al. [42] introduced a live streaming dataset in the gaming domain, but the content of the single-domain community culture is limited, making it difficult to extend the model to other domains. Additionally, the authors determined audience preferences solely based on the number of comments, with limited research on the rich intentions contained within the comments. Similar single-domain live streaming dataset works can be found in [3, 4, 18]. The problem of single-domain focus and task specificity in these works hinders the study of the rich semantics embedded in live streaming comments, leading to a research gap in domain-independent and in-depth understanding of comment information.

To address the aforementioned issues, we constructed a multimodal, multi-domain live streaming comment dataset **MMLSCU** and conducted annotations on the comments. We proposed four tasks related to comment understanding:

- Comment intent detection: Discerning the underlying intent of comments and identifying hidden intentions for a deeper understanding of users' thoughts and needs.
- Intent cause mining: Seeking to ascertain the rationale behind a specific intent, analyzing the deeper psychological factors that drive users to express certain intentions.
- Viewer comment explanation: Generating in-depth explanations of comments from the viewer's perspective, and breaking down barriers imposed by specific community cultures.

[1]https://www.twitch.tv

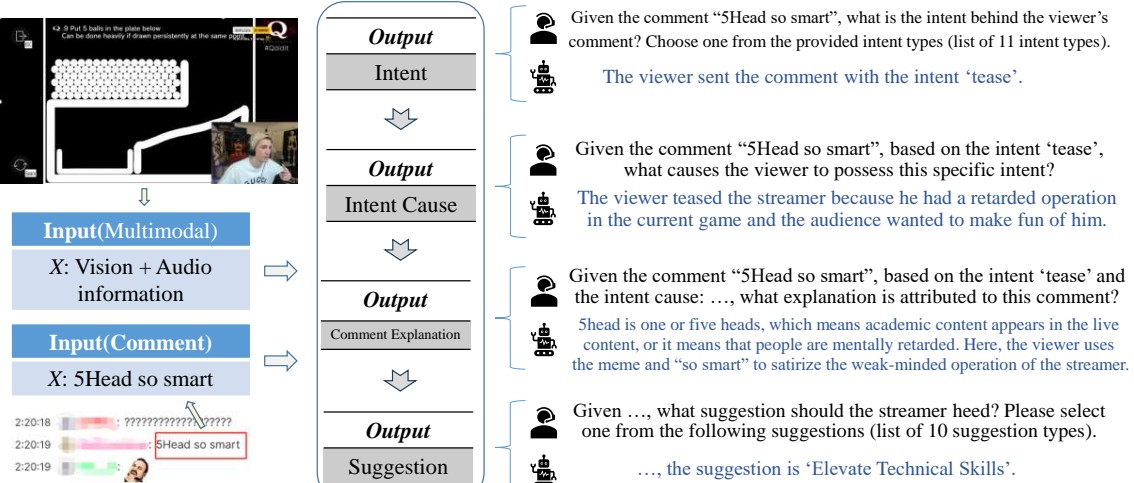

**Figure 1: On the left, we give an example to show the live streaming scenario including a streamer, viewers and their comments. On the right, we give four examples to show the tasks, namely comment intent detection, intent cause mining, viewer comment explanation and streamer policy suggestion, in our proposed MMLSCU dataset.**

- Streamer policy suggestion: Offering recommendations to streamers to help optimize content, adjust strategies, and increase user engagement.

Our dataset is sourced from Twitch live streams, encompassing video, audio, text, and emoji images. Twitch is a platform with a rich live streaming domain and a wide user base, making it a logical choice for dataset selection. Our dataset comprises 8 domains, totaling 200 live segments, selected from 150 streamers. Streamer selection was balanced across factors such as age and region. Specific statistical information is provided in Table 1.

For the four tasks we proposed, corresponding annotations were conducted. For task 1 and 4, we designed 11 intent labels and 10 suggestion labels respectively, and performed multi-person cross-annotations. For task 2 and 3, we created intent cause and comment explanation texts. Through these four tasks, we aim to achieve a fine-grained analysis of real-time live streaming comments, uncover the hidden insights within live comments. This will provide enriched feedback to streamers, enhancing the quality of their streams and ultimately driving progress in the industry.

The four tasks present certain challenges. On one hand, it requires a unified approach to handle four different forms of subtasks, and there are interdependencies between these tasks. The performance of the preceding task affects the subsequent task. On the other hand, each task necessitates the integration of multi-modal data over a period of time. For instance, due to cultural differences among different communities, only the video modality provides additional information to determine which community culture a comment belongs to, and thereby infer the intent of the comment.

To address the aforementioned challenges, we propose the Multi-Modal Four Comment Understanding Tasks (**MM⁴CU**) model, which consists of three components: **(1) Video Branch**, which encompasses a pre-trained visual encoder designed to extract features from video frames, a position embedding layer to infuse temporal information, and a video Q-former for consolidating frame-level

**Table 1: Statistics of the MMLSCU dataset.**

| Meta Item | Information |
|---|---|
| Total number of live streaming clips | 200 |
| Total number of comments | 50,129 |
| Total number of words in comments | 183,755 |
| Total number of emoticons in comments | 30,712 |
| Maximum number of words in comments | 65 |
| Average duration of live streaming clips(s) | 712.38 |
| Maximum duration of live streaming clips(s) | 840.00 |
| Average string length of comment texts | 21.70 |
| Maximum string length of comment texts | 500 |
| Total number of fields | 8 |
| Total nubner of streamers | 150 |

representations. **(2) Audio Branch**, which involves a pre-trained audio encoder, a position embedding layer to incorporate temporal information into audio segments, and an audio Q-former for integrating diverse audio segment features. **(3) Text Decoder**, this component, for the fused multi-modal information, constructs the chain-of-thought (CoT) [39] for prompt learning. In our proposed model, we leverage CoT to tackle these tasks step by step, revealing the inherent relationships among them. Additionally, by harnessing the robust generative capabilities of large-scale models, we use Video-LLaMA [43] as the foundational model to effectively integrate features from various modalities.

We conducted a series of experiments on our constructed MMLSCU dataset. Compared to the baseline, there were significant improvements in the F1 scores for both classification tasks: viewer comment intent detection and streamer policy suggestion. Additionally, the metrics for the two generation tasks, intent cause mining and viewer comment explanation, also surpassed baseline by a large margin. Furthermore, our ablation experiments indicated

that the introduction of multimodal information and CoT inference markedly enhanced the model's multi-step reasoning performance. The contributions of this paper include:

- We present a multi-modal, multi-domain live streaming comment dataset named MMLSCU. This dataset incorporates 4 distinct annotation types, facilitating the understanding of live streaming comments and furnishing streamers with nuanced feedback to elevate the overall streaming quality. To our knowledge, this work pioneers in filling this particular research void.
- Recognizing the associations of our proposed tasks, we architect a CoT framework to joint handle them and setup a strong benchmark for following-up work.
- Our empirical analyses on MMLSCU validate the superiority of utilizing multi-modal data compared to relying solely on text-based comment comprehension and feedback mechanisms.

**Our code and dataset are available at https://anonymous.4open.science/r/MM4CD-E683/ for reviewing.**

## 2 RELATED WORK

### 2.1 Live Streaming and Related Datasets

In recent years, live streaming has emerged as a pervasive phenomenon, particularly prominent within social media and the entertainment sector[35]. Empirical data suggests that most of the younger demographic has engaged with live streaming content at least once. Notably, several leading live streaming content creators have garnered a viewership that surpasses traditional television broadcasts[24]. This extensive viewership offers unparalleled opportunities for scholarly investigations into user engagement dynamics and intent recognition within the live streaming milieu. Furthermore, the evolving paradigms of digital gifting[19, 40] and bullet commentary, commonly referred to as "danmu"[14, 41], which are intrinsic to live streaming, present intriguing avenues for academic exploration.

Gaming-centric broadcasts are unequivocally recognized within the live streaming ecosystem as a predominant sub-domain. To facilitate a deeper understanding of user interactions within this context, a plethora of both unimodal[15, 16, 42] and multimodal datasets[3, 34, 36] have been curated. However, it is imperative to note that while gaming broadcasts occupy a pivotal position within the live streaming culture, they do not encapsulate its entirety. The spectrum of live streaming content is vast, encompassing domains such as casual conversations, educational sessions, culinary demonstrations, and travelogues, to name a few. These genres exhibit intrinsic disparities when juxtaposed with gaming broadcasts. Consequently, an exclusive reliance on datasets derived from gaming streams may not provide a holistic representation of the broader live streaming culture. While many unimodal datasets are tailored for the live streaming domain, notably those focusing on commentary text[1, 33], there is a discernible lacuna in the realm of comprehensive multimodal datasets. Several multimodal datasets are tailored for short video segments and bullet commentary annotations[11, 38]. However, datasets that offer a comprehensive multimodal perspective on live streaming remain relatively sparse. Chen[5] proposes that MovieLC Dataset, a multimodal dataset tailored for

the live streaming domain, is noteworthy. Yet, it predominantly aligns bullet commentaries with their corresponding video segments without delving into the underlying sentiment or the contextual triggers for such commentaries. Such nuanced information is pivotal for models aiming to better comprehend video content.

### 2.2 Multimodal Pretrained Models and CoT

The integration of textual and visual information has become a prominent research direction, leading to the emergence of several multimodal pretrained models. Building on the foundational success of unimodal architectures like BERT[6] for text and ResNet[13] for images, recent models aim to jointly learn representations across both modalities. Notably, CLIP[30] learns visual concepts from natural language supervision, demonstrating robust zero-shot performance across various visual benchmarks. Similarly, ViLBERT[23] employs a dual-stream approach, processing visual and textual inputs separately and then merging them, showcasing impressive results in visual question answering and visual commonsense reasoning.

Among these advancements, large-scale language models (LLMs), such as ChatGPT[27], stand out for their approach to human-level intelligence[31]. VideoLLM[43] distinguishes itself by adeptly integrating visual and textual information from videos, emphasizing the narrative structure and temporal dynamics, proving its efficacy in tasks requiring a profound understanding of video content. Furthermore, there's compelling evidence that LLMs possess an exceptional aptitude for common-sense understanding[22, 29].

Transitioning from their intrinsic understanding abilities, the introduction of CoT technique has gained prominence[44]. CoT has been widely used to enhance the multi-step reasoning capabilities of LLMs by encouraging them to generate intermediate reasoning chains, guiding them towards problem solutions[39]. Notably, CoT prompting is a gradient-free approach that coaxes these models into articulating the intermediate steps leading to the final answer.

## 3 THE MMLSCU DATASET

The overall process of constructing the MMLSCU dataset is illustrated in Fig 2. In this section, we will introduce the details of data preparation, task definitions and data annotation, respectively.

### 3.1 Data Preparation

Due to the high quality, diversity, and wide viewer appeal of Twitch live streaming, we have selected Twitch as our data source for live streaming research. We enlisted the services of 14 seasoned viewers, well-versed in diverse online streaming scenarios, to observe live broadcasts or replays on Twitch for over a month. These scenarios include 8 domains: **Games**, **Just Chatting**, **In Real Life**, **Music and Performances**, **E-sports**, **Creative and Arts**, **Education and Learning**, and **Special Events**, In our dataset, 200 English live streaming clips since 2020 were selected, across total 2374 minutes and accompanied by 50,129 comments. The content of these clips is described and recorded at intervals of 20 seconds. To ensure quality content and meanwhile keep diversity, the data selection strategy considers various factors such as streamer age, gender, and viewer comment count. Pertinent factors are stored in

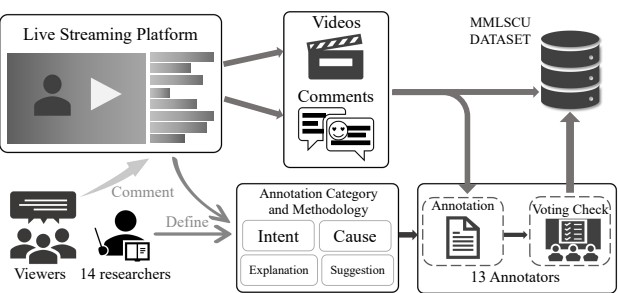

**Figure 2: The overall process of constructing the MMLSCU dataset.**

the meta-information corresponding to each live streaming clip in the dataset.

**Ethical Consideration** We utilized the Twitch Developer API[2], to obtain live streaming clips, strictly adhering to fair use principles. As per Section 8 of Twitch's terms of service[3], viewer comments (live chat during streaming) are unlicensed, permitting us to process them. After data acquisition, any sensitive or unsavory comment information was removed. Furthermore, to protect privacy information, all data underwent de-identification, removing viewer identity information irrelevant to live streaming content.

## 3.2 Task Definitions

*3.2.1 **Comment Intent Detection**.* We devised a novel multimodal live streaming comment intent detection classification method, encompassing 4 coarse-grained and 11 fine-grained intent categories. The 4 coarse-grained intent types includes **Achieve Goals**, **Express Emotions**, **Platform Operational Interactions** and **Another**. We randomly sampled each domain, using coarse-grained intent types for preliminary labeling. We discovered that the coarse-grained intent labels were overly broad when describing intents in complex live-streaming scenarios. Consequently, we refined each coarse-grained type into finer granularities, pre-labeled the randomly sampled data, and merged labels with ambiguous distinctions, resulting in 11 fine-grained label types, including like, unlike, hope, questioning, tease, express_surprise, express_abashed, normal_interaction, meme, none. The examples and detailed explanations of these labels can be found in Appendix A.1.1.

*3.2.2 **Intent Cause Mining**.* Relying solely on intent labels is insufficient to comprehensively describe the motivations and psychological states behind the viewer's comments. Therefore, we proposed a novel task to analyze the intent cause behind a comment, revealing the reasons prompting viewers to make specific comments in the prevailing live streaming context, facilitating the comprehension of semantic significance conveyed in associated comments. Due to the non-explicit nature of intent behind comments in live streaming, "intent cause" most of the time can't be extracted directly from comments, and mining "intent cause" requires consideration of the live streaming scenario relevant to the current comment. We utilize a generative approach to obtain the "intent

[2]https://dev.twitch.tv/docs
[3]https://www.twitch.tv/p/en/legal/terms-of-service/

cause" and regarding this task, we have manually written "intent cause" based on the comments and their relevant live streaming scenarios as the label data. Considering the enormity of manually crafting the causes for intents, we leveraged GPT4 to autonomously generate "intent cause" for scenariosn The generated cause was subsequently filtered based on the criteria shown in the Appendix A.4: Among the final annotations, we successfully generated approximately 30% of "intent cause" using GPT4, satisfying the above criteria.

Another common scenario is that the cause is not directly available from the comments and requires information from the multimodal live streaming scene to obtain it. This scenario would require annotators to write a reason by hand, and GPT4 would give the wrong cases, as shown in the Appendix A.4.

*3.2.3 **Viewer Comment Explanation**.* Comment explanation is devised to delve deeply into the inherent meanings behind comments, interpreting not merely the detailed meaning of sentences but also amalgamating the prevailing live streaming context to interpret from the viewer's perspective holistically. Let the comment explanation $CE$ be defined as the amalgamation of the following two components: $CE = SI + CI$. Herein, $SI$ denotes the sentence's intrinsic meaning, While $SI$ is autonomously generated by GPT4, it remains incumbent to manually sift through, especially for the latest internet slang, which GPT4 might not interpret accurately. As per our finalized annotations, approximately 90% of $SI$ could be autonomously generated via GPT4, satisfying the filtering criteria in the Appendix A.4. On the other hand, $CI$ captures the specific significance of sentences within the live streaming, highlighting the information the viewer intended to convey while commenting, engendering a profound comprehension of latent meanings. Relevant examples can be located in the Appendix A.4.

*3.2.4 **Streamer Policy Suggestion**.* Upon a profound understanding and analysis of live streaming comments, furnishing precise suggestions to streamers is a pivotal step to harnessing viewer feedback judiciously. suggestions were taxonomically classified into five primary categories: **Content Strategy**, **Engagement Strategy**, **Streaming Ethics**, **Streaming Environment**, **Others**. Acknowledging that a broad categorization might prove insufficient to distinguish the multitude of scenarios within a live streaming context, we further subdivide the suggestion types into 11 distinct categories, including switch up streaming content, elevate technical skills, boost audience interaction, avoid live streaming conflicts, enhance streaming conditions, resolve streaming errors, be mindful of words and actions, improve streaming attitude, keep up the good show, None. The detailed explanations of these labels can be found in Appendix A.5.1. A period corresponds to a continuous live streaming scene, and representative suggestions exist for this continuous live streaming scene. For each comment over a period of time, we combine the live streaming content to determine whether the suggestion type of the comment is a representative suggestion. If so, the suggestion type is marked as a representative suggestion. Otherwise, it is marked as None. Since the content of comments is various, if the suggestion type of each comment is considered, the streamer will get different suggestion types in a live streaming scene rather than the most typical suggestion type most needed in this scene.

## 3.3 Data Annotation

Following the data preparation and annotation definitions, we engaged 13 personnel, all fervent and adept with the live streaming environment, to undertake the task of data annotation. Staff members were equipped with exemplary instances for each annotation type to serve as guiding benchmarks. Only those who underwent comprehensive training were permitted to annotate. To amplify the efficiency of the annotation process, we established a dedicated database to manage all multi-modal data and a user-friendly annotation interface. The team was bifurcated into two distinct units: an eight-member annotation team and a five-member review panel. The annotation team collaboratively labeled the entire dataset with predefined labels for intent and suggestion types, choosing the most fitting intent and suggestion type. Subsequently, the review panel vetted these preliminary labels through a voting mechanism. If a label garnered acceptance from three or more reviewers (a 3-out-of-5 majority), it was ratified; otherwise, it was earmarked for re-annotation until achieving the requisite majority acceptance. For annotations necessitating manual writing or generation through GPT4, one member operated GPT4 for generation, while seven undertook manual scripting. The five-member review panel also employed the voting mechanism to ensure quality. When annotating emoticons, referencing external platforms, such as Know Your Meme[4], enhancing annotation precision.

In Appendix A.2, We provide specific distribution details for two classification task labels and statistics regarding the number of comments in different domains.

## 4 METHODOLOGY

Utilizing a multi-modal form of large language models, we engineered a CoT framework expressly tailored for the tasks we had delineated.

## 4.1 Model Architecture

According to literature focusing on observing live E-sports games on the Twitch platform, it becomes imperative to account for the time spectators invest in crafting their comments while watching the live streaming. This consideration arises due to inherent delays, such as typing time, which imply that comments posted by viewers at a given instance frequently pertain to live streaming content a few moments prior. In a user study documented by Palin[28], the average typing speed on keyboards, denoted as $WPM_k$, is found to be 52 words per minute, while the average typing speed on smartphones, referred to as $WPM_s$, is 38 words per minute. Incorporating both the $WPM_k$ and $WPM_s$ metrics to obtain the time of live streaming content related to the current comment. For any current comment, we obtain its time as $T_c$. Given that viewers cannot foresee forthcoming live scenarios, we consider the start time, $T_s$, of the segment of live streaming content corresponding to that particular comment. Define $l$ as the prior duration of live streaming content associated with the current comment. $l$ can be determined using the given formula:

$$l = n_w \Bigg/ \frac{0.5 * (WPM_k + WPM_s)}{60}, \qquad (1)$$

where $n_w$ is the number of words in the current comment. The formula of $T_s$ is as follows:

$$T_s = MAX(T_c - l, 0). \qquad (2)$$

Therefore, considering a comment timestamped at $T_c$, the corresponding video and audio modalities should encompass live streaming content within the time frame $(T_s, T_c)$. We interpret the image as a single video frame; thus, the emoticon $Emote_{T_c}$ in a comment $Comment_{T_c}$ at time $T_c$ is treated as one video frame concatenated after the video frames within $(T_s, T_c)$. Subsequently, the video and audio representations are relayed to the Language Learning Model (LLM) to align with the text embeddings' dimensions. The Vision branch processes the video and derives its representation, while the Audio branch is utilized for audio representation.

**Vision Branch** The objective of the Vision branch is to facilitate the comprehension of visual input by the LLM. This branch encompasses a pre-trained visual encoder designed to extract features from video frames, a position embedding layer to infuse temporal information within the frames, a video Q-former to aggregate frame-level representations, and a linear layer tasked with projecting these video outputs to a dimension congruent with the LLM's text embeddings. For the visual encoding process, we incorporate the pre-trained visual component of BLIP-2 [20] as the frozen visual encoder, which includes a ViT-G/14 [7] from EVA-CLIP and a pre-trained Q-former. The position embedding layer, video Q-former, and linear layer are initialized randomly and fine-tuned to effectively bridge the output of the frozen visual encoder with that of the frozen LLM.

**Audio Branch** The Audio branch is constructed to enable the LLM to interpret audio inputs. It comprises a pre-trained audio encoder, a position embedding layer to embed temporal information into audio segments, an audio Q-former to amalgamate different audio segment features and a final linear layer to map the audio representation to the embedding space of the LLM. We employ the pre-trained Imagebind [8] as our audio encoder. Analogous to the video Q-former, the audio Q-former instills temporal information by appending learnable position embeddings to audio segments. It subsequently generates fixed-length audio features by calculating interactions between position-encoded audio segments. The architecture of the audio Q-former mirrors that of the video Q-former, and, ultimately, a linear layer maps these audio features to the embedding domain of the LLM.

## 4.2 Chain-of-Thought Prompting

Considering the interrelations among the four tasks, we designed a multi-modal version of the CoT framework. This framework encompasses four sequential phases, specifically tailored to handle the tasks: viewer comment intent detection > intent cause mining > viewer comment explanation > streamer policy suggestion. While all four stages employ a consistent model architecture, variations are introduced in the input and the output.

For a comment $Comment_{T_c}$ at timestamp $T_c$, its corresponding live streaming content is during the $(T_s, T_c)$ interval. $Comment_{T_c}$ consists of $Emote_{T_c}$ and $Text_{T_c}$, where $Emote_{T_c}$ can be empty.

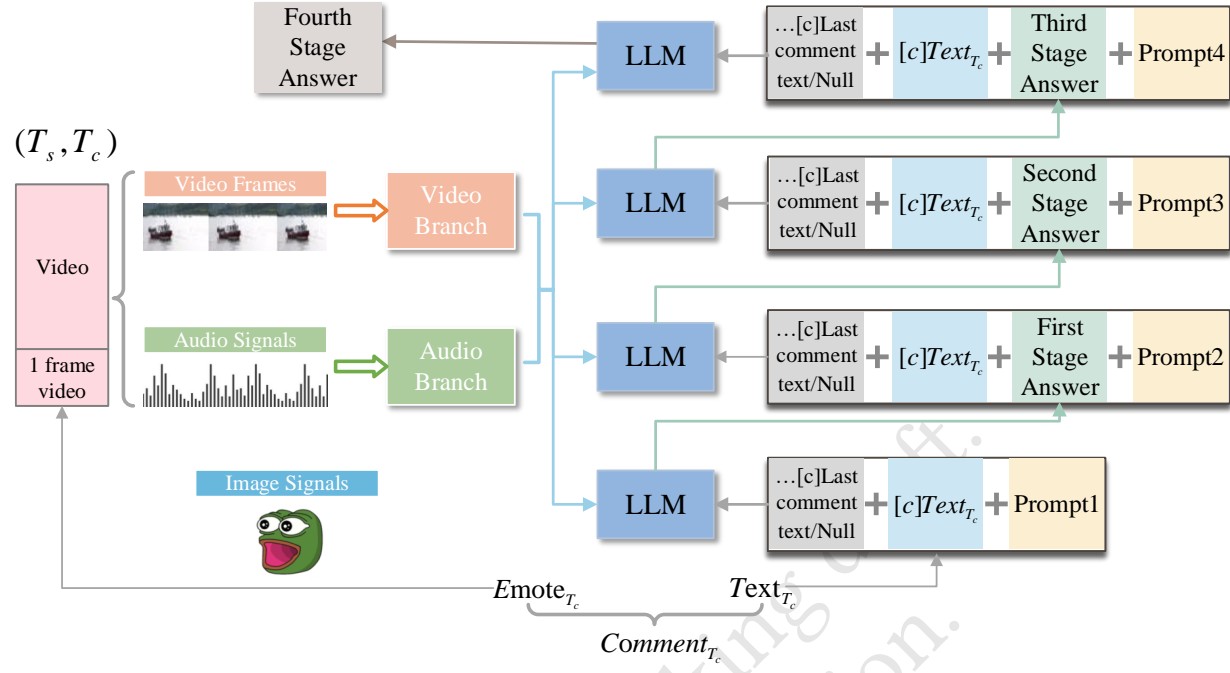

**Figure 3: The architecture of our MM$^4$CU model for joint multimodal live streaming comment understanding with CoT schema.**

**First Stage** In the first stage, our input is represented as

$$X^1 = \{x^1_{text}, x_{vision}, x_{audio}\}, \qquad (3)$$

where:

$$x^1_{text} = x^1_{last} \circ Text_{T_c} \circ Prompt^1, \qquad (4)$$

- $x^1_{text}$ denotes the text part of input in first stage.
- $x_{vision}$ denotes the vision part of input.
- $x_{audio}$ denotes the audio part of input.
- $x^1_{last}$ denotes the text part of the comments preceding the current comment in the first stage.
- $\circ$ denotes the *concatenate* operation.
- $Text_{T_c}$ denotes the text component of the current comment.
- $Prompt^1$ denotes the prompt context of the first stage.

When the current comment is inaugural, $x^1_{last}$ defaults to $NULL$. Otherwise, comments are demarcated by the delimiter $[C]$. If appending the text from a prior comment exceeds the maximum input length, the over-extending portion of that comment's text is truncated. In the first stage, The prompt template is articulated as:

> **Template − Stage 1**
>
> $C_1$[Given the comment $Text_{T_c}$ and its accompanying multi-modal live streaming data], what is the intent behind the viewer's comment? Choose one from the provided intent types (list of 11 intent types).

Here $C_1$ signifies the prompting context for the first stage. This can be formally expressed as $I=\text{argmax}\,p(i|Text_{T_c})$, where $I$ represents the output text denoting the comment's intent, also visualized as the *First Stage Answer* in the figure 3.

**Second Stage** In the second stage, our input is represented as

$$X^2 = \{x^2_{text}, x_{vision}, x_{audio}\}, \qquad (5)$$

where:

$$x^2_{text} = x^2_{last} \circ Text_{T_c} \circ First\ Stage\ Answer \circ Prompt^2, \quad (6)$$

In the second stage, The prompt template is articulated as:

> **Template − Stage 2**
>
> $C_2[C_1, I]$. Based on the identified intent, what causes the viewer to possess this specific intent?.

$C_2$ acts as the prompting context for the second stage, concatenating $C_1$ and $I$. Mathematically, this is represented by $C=\text{argmax}\,p(c|Text_{T_c}, i)$, where $C$ is the textual answer encapsulating potential causes for the intent, or as illustrated in the figure 3, *Second Stage Answer*.

**Third Stage** In the third stage, our input is represented as

$$X^3 = \{x^3_{text}, x_{vision}, x_{audio}\}, \qquad (7)$$

where:

$$x^3_{text} = x^3_{last} \circ Text_{T_c} \circ Second\ Stage\ Answer \circ Prompt^3. \quad (8)$$

In the third stage, The prompt template is articulated as:

> **Template − Stage 3**
>
> $C_3[C_2, C]$. Grounded on the cause of intent, what explanation can be attributed to this comment?

**Table 2: The experimental results for the "comment intent detection" and "streamer policy suggestion" tasks on the MMLSCU test set.** $Acc_{11}$ **represents the "comment intent detection" task that has 11 labels, and** $Acc_{10}$ **represents the "streamer policy suggestion" task that has 10 labels.**

|  | $Acc_{11}$ | P | R | $F_1$ | $Acc_{10}$ | P | R | $F_1$ |
|---|---|---|---|---|---|---|---|---|
| MAG-BERT | 66.31 | 65.83 | 63.46 | 64.62 | 60.15 | 60.65 | 55.46 | 57.94 |
| MUIT | 64.59 | 62.48 | 66.24 | 64.31 | 59.98 | 58.03 | 56.07 | 57.03 |
| MISA | 65.72 | 63.05 | 65.57 | 64.29 | 59.04 | 57.73 | 54.80 | 56.23 |
| MM$^4$CU | 73.00 | 75.23 | 71.59 | 73.36 | 71.06 | 71.32 | 69.48 | 70.39 |

**Table 3: The experimental results for the "intent cause mining" and "viewer comment explanation" tasks on the MMLSCU test set.**

|  | Intent Cause Mining | | | | Viewer Comment Explanation | | | |
|---|---|---|---|---|---|---|---|---|
|  | $B^3$ | $B^4$ | ROUGE-L | METEOR | $B^3$ | $B^4$ | ROUGE-L | METEOR |
| UniVL | 18.23 | 12.61 | 22.15 | 23.76 | 15.32 | 10.11 | 19.74 | 22.35 |
| MM$^4$CU | 33.10 | 27.15 | 37.98 | 34.05 | 31.21 | 26.12 | 35.93 | 30.23 |

$C_3$ is the prompt context for the third stage, concatenating $C_2$ and $C$. This is mathematically framed as $E=\text{argmax}\,p(e|Text_{T_c}, i, c)$, wherein $E$ is the text capturing potential explanations of the comment. $E$ is termed the *Third Stage Answer* in the figure 3.

**Fourth Stage:** In the fourth stage, our input is represented as

$$X^4 = \{x^4_{text}, x_{vision}, x_{audio}\}, \qquad (9)$$

where:

$$x^4_{text} = x^4_{last} \circ Text_{T_c} \circ Third\ Stage\ Answer \circ Prompt^4. \qquad (10)$$

In the Fourth stage, The prompt template is articulated as:

> **Template − Stage 4**
>
> $C_4[C_3, E]$. Based on the comment explanation, what suggestion should the streamer heed? Please select one from the following suggestions (list of 10 suggestion types).

$C_4$ operates as the prompt context for the fourth stage, concatenating $C_3$ and $E$. $S=\text{argmax}\,p(S|Text_{T_c}, i, c, e)$, wherein $S$ denotes the resultant suggestion text or the *Fourth Stage Answer* illustrated in the figure 3.

### 4.3 Training Strategy

For the pre-training of the Vision branch, we employed our live-stream clips along with their associated descriptive metadata. we incorporated a video-to-text generative task, wherein a 20-second live-stream video and its corresponding description served as inputs to prompt the frozen LLM to generate an apt text description. The objective of this phase was to leverage live-stream data to imbue the video features with as much live-stream scenario knowledge as possible. Given the scarcity of audio-text data, directly training the Audio branch posed significant challenges. The aim of the learnable parameters within the audio-language branch was to

align the output embedding of the frozen audio encoder with the LLM's embedding space. After pre-training, our model was fine-tuned using our annotated dataset.

## 5 EXPERIMENTS

In this research, we propose a multimodal dataset designed to provide a robust foundation for studies in the field of live streaming. Upon finalizing our dataset, we proceeded to segregate it into training, validation, and test sets in an 8:1:1 ratio. To evaluate the effectiveness of our dataset, we conducted a series of experiments and compared the results with existing models. This section will introduce our experimental setup and the analysis of the experiments.

### 5.1 Experimental Setup

**Baseline** To assess the performance of existing methods on our dataset, we conducted a series of experiments. For the intent detection and policy suggestion tasks, we selected the following models: **MAG-BERT** [32], **MulT** [37], and **MISA** [12]. For the intent cause mining and viewer comment explanation tasks, which are two generative tasks, we conducted experiments using the multimodal generative model **UniVL** [25].

**Evaluation Metrics** We used various evaluation metrics to assess the model performance. For the classification tasks, following the MuIT [37] framework , we reported n-class accuracy ($Acc_{11}$ for intent detection score classification, $Acc_{10}$ for policy suggestion score classification), $F_1$ score, precision (P), and recall (R), calculated using macro-averaging[9]. For the generative tasks, we reported evaluation metrics such as $B^3$ and $B^4$, ROUGE-L[21], METEOR[2].

**Training Details** We pre-trained our model on 2 NVIDIA Tesla A100 GPUs. We employed a learning rate warm-up strategy[10], starting with an initial learning rate of 0.0001 and linearly increasing it to 0.001, after which it remained constant. The batch size was

**Table 4: The ablation studies for the "comment intent detection" and "streamer policy suggestion" tasks on the MMLSCU test set.** $Acc_{11}$ **represents the "comment intent detection" task,** $Acc_{10}$ **represents the "streamer policy suggestion" task. "-" means reducing of the condition.**

| | $Acc_{11}$ | P | R | $F_1$ | $Acc_{10}$ | P | R | $F_1$ |
|---|---|---|---|---|---|---|---|---|
| MM$^4$CU | 73.00 | 75.23 | 71.59 | 73.36 | 71.06 | 71.32 | 69.48 | 70.39 |
| only text | 71.89 | 73.98 | 70.14 | 72.01 | 69.17 | 69.82 | 67.54 | 68.66 |
| - Vision | 72.14 | 74.61 | 70.82 | 72.67 | 70.32 | 70.65 | 67.83 | 69.21 |
| - Audio | 72.61 | 74.83 | 71.11 | 72.92 | 70.66 | 70.90 | 68.75 | 69.81 |
| - CoT | 73.00 | 75.23 | 71.59 | 73.36 | 66.30 | 67.77 | 63.09 | 65.35 |

**Table 5: The ablation studies for the "intent cause mining" and "viewer comment explanation" tasks on the MMLSCU test set.**

| | Intent Cause Mining | | | | Viewer Comment Explanation | | | |
|---|---|---|---|---|---|---|---|---|
| | $B^3$ | $B^4$ | ROUGE-L | METEOR | $B^3$ | $B^4$ | ROUGE-L | METEOR |
| MM$^4$CU | 33.10 | 27.15 | 37.98 | 34.05 | 31.21 | 26.12 | 35.93 | 30.23 |
| only text | 31.23 | 25.38 | 35.42 | 32.17 | 29.40 | 24.78 | 33.76 | 28.17 |
| - Vision | 32.01 | 26.12 | 36.91 | 33.09 | 29.97 | 25.06 | 34.50 | 29.11 |
| - Audio | 32.60 | 26.65 | 37.48 | 33.55 | 30.71 | 25.76 | 34.03 | 28.65 |
| - CoT | 30.89 | 23.40 | 33.65 | 31.04 | 26.96 | 23.22 | 31.34 | 27.79 |

set to 64, and we used the Adam optimizer[17] with β1 set to 0.9 and β2 set to 0.999. We iterated for a total of 10 epochs.

## 5.2 Main Result

The experimental results for the classification tasks are shown in Table 2 (all results are macro-averaged values). The results of the generation task are shown in Table 3.

From the experimental results, it is evident that our model has achieved a significant improvement compared to the baseline models. This improvement can be attributed to our innovative multi-modal architecture and the powerful inferential capabilities of the large text model. Furthermore, it can be observed that the classification performance for the "streamer policy suggestion" task is lower than that of the "comment intent detection" task. This is because the streamer policy suggestion is our fourth task, and its results are influenced by the outcomes of the preceding three tasks. Additionally, making policy suggestions requires the synthesis of information from a previous time period, making it a more challenging task compared to intent classification, hence resulting in lower performance metrics.

## 5.3 Ablation Studies

In assessing the influence of different modal information on classification and generation tasks, we carried out additional ablation experiments. The results of these experiments are detailed in Table 4 and Table 5, for classification and generation tasks respectively.

From the experimental results, it can be observed that removing either video or audio information results in a slight decrease in model performance. The drop in performance is slightly more

pronounced when video information is removed compared to audio information. This is because in live streaming scenarios, audience comments are often responses to the actions of the streamer. Furthermore, in the case where only the text modality is available, both classification tasks see a decrease of 1.11 and 1.89 in accuracy and a decrease of 1.35 and 1.73 in F1 score, respectively. The generative task metrics also show some decline, indicating that video and audio modalities indeed provide essential information for the classification tasks.

Additionally, as the intent classification task serves as the first step in our reasoning process, removing the CoT doesn't affect the model's performance. However, the subsequent three tasks rely on the CoT provided in the previous step, and removing the CoT results in a significant performance drop. This further underscores the importance of CoT in the multi-step reasoning process.

## 6 CONCLUSION

Our research has created MMLSCU, a multimodal and cross-domain live streaming comment dataset, along with four comment understanding tasks. These tasks include comment intent detection, intent reason mining, audience comment explanation, and broadcaster strategy recommendations. Through experimentation, we have demonstrated that the introduction of multimodal data and CoT reasoning significantly improves model performance. This research fills a gap in domain-independent and in-depth comment information understanding, providing essential tools for enhancing live streaming quality and driving industry development. We will openly share our dataset and code to encourage more researchers to participate in future studies and further advance this field.

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

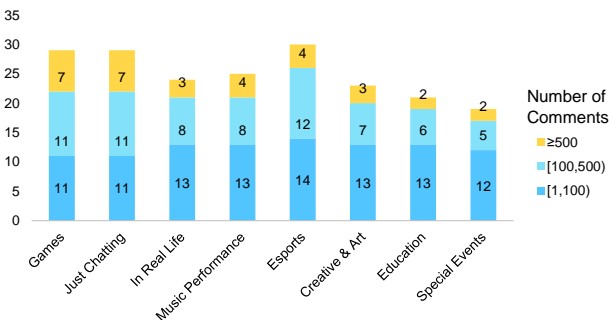

**Figure 4: Distribution of domain and number of comments in live slices.**

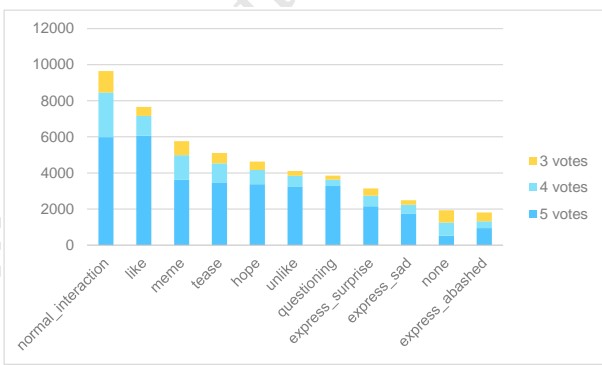

**Figure 5: Intent type distribution and voting results**

## A DATASET DETAILS

### A.1 Label Explanation

*A.1.1 Label Explanation for Comment Intent Detection Task.* The labels explanation for comment intent detection task are in Table 6.

### A.2 Statistical Information

To further explore the dataset's domain characteristics and the distribution of labels for the two classification tasks, in Figure 4, we've provided a breakdown of comment counts across various live streaming domains. In Figure 5, we've presented the vote counts for different intent labels during the annotation process, and in Figure 6, we've shown the vote counts for various recommendation labels.

### A.3 Prompt Template for Labeling Task

For intent cause mining and viewer comment explanation, we created the following prompt templates to generate labels.

> *Template*
>
> Given a live streaming comment with the content [*comment*] and an associated intent of [*intent*], what is the cause behind this intent?

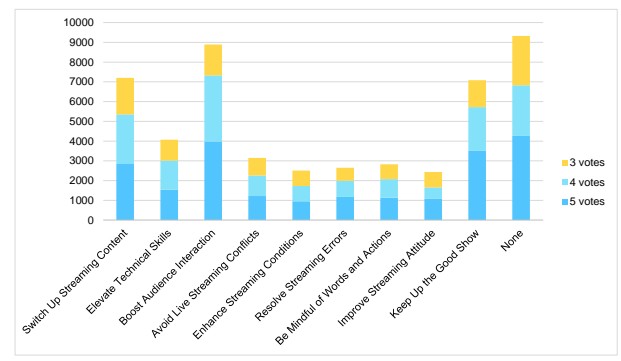

**Figure 6: Suggestion type distribution and voting results**

> *Template*
>
> Given a live streaming comment with the content [*comment*] , what is the explanation of the intrinsic meaning of the comment?

## A.4 GPT4 Filtering Criteria and Generation Cases

**GPT4 Filtering Criteria**:

Relevance - the generated cause must exhibit a strong correlation with both the comment's content and its intent; Logical Consistency - the cause should align logically with the live streaming context and content; and Conciseness - the described cause should be succinct, eschewing undue complexity.

**Wrong Generation Cases**:

Comment: *This music takes me back!*

Intent: Express_sad

Task: Intent Cause Mining

GPT4's Response: *The user might be reminded of a popular dance trend from a few years ago due to the music.*

Error analysis: In a live streaming scene in the domain of music, the streamer sang a very touching old song that could make the viewer cry, evoking the viewer's sad feelings of missing the past and feeling that time has passed and things have changed. The comments of the live streaming were all expressing such sadness, but the answers generated by GPT4 were obtained only by comments. It represents a connection to trend or happiness rather than sadness or melancholy in a live streaming scene.

**Acceptable Generation Case**:

Comment: *That technique is straight out of the 90s!*

Task: Viewer Comment Explanation.

SI: *The technique being demonstrated or discussed is reminiscent of outdated technology or things from the 1990s.*

Live streaming scenario: As shown in the figure 7

CI: *The viewer believes that the product appears dated and similar to older technique from the 1990s. This hints at a criticism that this product is not as advanced as streamer might think and is not worth it.*

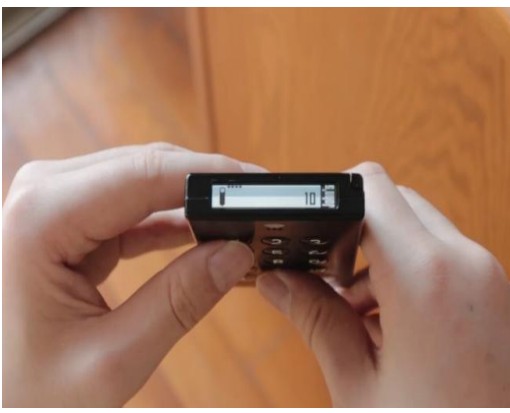

**Figure 7: The streamer is showcasing a newly purchased electronic products.**

CE: *The technique being demonstrated or discussed is reminiscent of outdated technology or things from the 1990s. The commenter believes that the product appears dated and similar to older technique from the 1990s. This hints at a criticism that this product is not as advanced as streamer might think and is not worth it.*

## A.5 Dataset File Structure

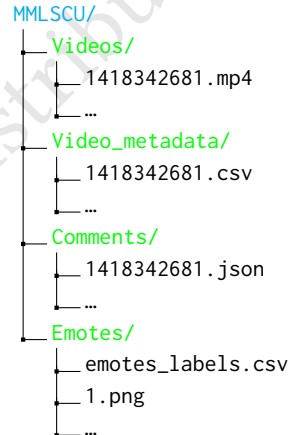

*A.5.1 Label Explanation for Streamer Policy Suggestion Task.* The labels explanation for streamer policy suggestion task are in Table 7.

**Table 6: The explanation for comment intent detection task's labels.**

| | Intent | Explanation |
|---|---|---|
| Achieve Goals | like | Like, Support, Enjoy, Comfortable and Pleasant |
| | unlike | Dislike, Oppose, Threaten, Irrational |
| | hope | Hope, Suggestion, Spectator |
| | questioning | Question, Doubt, Confusion |
| | tease | Mock, Ridicule |
| Express Emotions | express_surprise | Expressing surprise or astonishment |
| | express_sad | Expressing sadness or regret |
| | express_abashed | Expressing awkwardness or embarrassment |
| Platform Operational Interactons Another | normal_interactoin | The normal interaction in a livestream room |
| | meme | Silly antics and meme play in the livestream room |
| | none | No comment posted or unclear comment intent |

**Table 7: The explanation for comment intent detection task's labels.**

| | Suggestion | Explanation |
|---|---|---|
| ContentStrategy | Switch Up Streaming Content | The audience finds the current livestream content too dull and suggests switching to or trying out new content. |
| | Elevate Technical Skills | The audience thinks the streamer is not skilled enough and suggests that the streamer should improve their technical proficiency. |
| EngagementStrategy | Boost Audience Interaction | The audience feels that the streamer lacks interaction with them, verlooks their opinions and requests, and suggests that the streamer should enhance interaction with the audience. |
| | Avoid Live Streaming Conflicts | The streamer or certain audience members in this livestream room are engaging in provocative or conflict-inducing behavior. It is advised that the streamer takes steps to avoid conflicts during the livestream. |
| StreamingEnvironment | Enhance Streaming Conditions | The audience feels that the streamer's livestream equipment conditions are subpar, or there are issues with background noise. They suggest that the streamer should improve the livestreaming environment. |
| | Resolve Streaming Errors | The streamer is currently experiencing issues with the livestream, such as network problems or a disabled camera.It is suggested that the streamer promptly address and resolve these errors. |
| StreamingEthics | Be Mindful of Words and Actions | The audience is warning the streamer about discussing orengaging in inappropriate topics or actions, such as skirting the edges or promoting racial discrimination. They emphasize the importance of the streamer being mindful of their words and actions. |
| | Improve Streaming Attitude | The audience feels that the streamer is not putting enough effort into the livestream and seems distracted. They suggest that the streamer should livestream with more dedication and a focused mindset, and correct their attitude. |
| | Keep Up the Good Show | The audience is very satisfied with the current program and encourages the streamer to keep up the good work. They hope the streamer will continue to maintain this level of performance. |
| | None | |

