# OpenReview forum: "MMLSCU: A Dataset for Multi-modal Multi-domain Live Streaming Comment Understanding"
_ACM.org/TheWebConf/2024/Conference — TheWebConf24 Oral_

### Official Review · Reviewer_JaJW · 2023-11-06

**Novelty:** 5
**Technical Quality:** 4

**Review:**

This paper is based on the online live streaming scenario and aims to utilize large language models to integrate videos, audios from live streaming, and comments from users. The goal is to analyze the intentions behind these live streaming comments and help improve the quality of live streaming for broadcasters. To achieve this purpose, the authors collected and created a multimodal, multi-domain live streaming comment dataset. They implemented an end-to-end model using CoT (Chain of Thought) that sequentially performs tasks such as audience comment intent detection, intent cause mining, audience comment explanation, and streamer policy suggestion based on this dataset.

**Strengths:**

* This paper explored a novel problem, which is the mining of user comments in the context of online live streaming. The authors collected and annotated a new dataset for this problem and open-sourced both the dataset and the code, which has contributed to the advancement of research in this field to some extent.
* The proposed method integrated multimodal information such as video, audio, and text. It utilized pre-trained encoders for each modality and employed the Chain of Thought (COT) framework, centering around large language models, to perform reasoning for various tasks.
* Experiments demonstrated the advantages of the proposed method in mining live streaming comments by fusing multimodal information compared to approaches using fewer modalities.
* The structure of this paper is clear, and the illustrations make it easy for readers to understand the paper's objectives and the proposed methods.

**Weaknesses:**

* In my understanding, the authors only utilized the content of comments (without accessing video and audio information from the live stream) to generate viewer comment explanations using GPT-4. Even with subsequent manual filtering, I think the explanations obtained might still be somewhat unreliable.
* The annotation for streamer policy suggestions, in my opinion, is highly subjective and may not accurately represent the genuine suggestions from the commenters. For instance, a viewer teasing an entertainment streamer's gaming behavior might just be expressing their emotions to other viewers, rather than giving a specific suggestion about the streamer's technical skills. Perhaps this viewer watches the stream specifically for the entertainment derived from the streamer's relatively poor technical skills. Besides the filtering through multi-person voting, are there any other criteria or guidelines used for annotating streamer policy suggestions?
* The process of obtaining the final suggestions through multi-turn conversations requires multiple executions of Large Language Model (LLM) inference, which is detrimental to the overall efficiency of the model.

**Questions:**

* Could the authors provide more detailed information about the dataset annotation process?
* Can the proposed multi-turn COT process be optimized into a single-turn approach to reduce the computational cost of inference?
* How is it determined whether a user is providing suggestions to the streamer or engaging in regular conversation with other viewers or the streamer?

**Reviewer Confidence:**

4: The reviewer is certain that the evaluation is correct and very familiar with the relevant literature

**Scope:**

3: The work is somewhat relevant to the Web and to the track, and is of narrow interest to a sub-community

---

### Official Review · Reviewer_mL4w · 2023-11-24

**Novelty:** 4
**Technical Quality:** 4

**Review:**

This paper proposes a multi-modal and multi-domain live streaming dataset. The dataset contains 4 annotation types and provides important data support for research of live streaming industry.
Clarity:
The paper gives a clear presentation of the proposed dataset and method.
Originality: The four defined tasks and the multi-modal CoT framework, adds to its Originality.
Pros:
The dataset contains 4 annotation types and provides important data support for research of live streaming industry.
The multi-modal CoT framework is novel and well-designed which setup a benchmark for following-up work.
Cons.
The dataset only contains 200 clips, may not be enough to achieve a fine-grained analysis.
It may be helpful to include the running time since four stages CoT is conducted.
In table 1, “Total nubner of streamers” -> “Total number of streamers”

**Questions:**

•	The section 3.3 provides details of data annotation of two classification task, how the comment explanation and streamer policy suggestion annotated?
•	How the model is finetuning on the annotated dataset? More details should be included.

**Reviewer Confidence:**

4: The reviewer is certain that the evaluation is correct and very familiar with the relevant literature

**Scope:**

3: The work is somewhat relevant to the Web and to the track, and is of narrow interest to a sub-community

---

### Official Review · Reviewer_Ccag · 2023-11-24

**Novelty:** 6
**Technical Quality:** 5

**Review:**

Pros:

1. The paper is well-written and easy to follow.

2. The authors present a novel multi-modal, multi-domain live streaming comment dataset named MMLSCU and propose four research tasks on this dataset, which fills a research gap in this area.

3. The paper introduces a CoT framework to jointly handle the proposed tasks and sets up a strong benchmark for future work.

4. Empirical analyses on MMLSCU demonstrate the superiority of utilizing multi-modal data compared to relying solely on text-based comment comprehension and feedback mechanisms.

Cons:

1. While the authors provide details on the prompt construction for the four stages, some important details are missing in the paper. For example, it is unclear which specific LLM is employed in the MM4CU model, and whether different LLMs were tested as the backbone of the model.

2. The experimental comparison in Section 5.2 seems insufficient and potentially unfair. The introduction to the baseline methods in Section 5.1 is too brief, and the sizes of these models appear to be much smaller than the proposed model since it contains an LLM.

3. In Section 5.3, the authors conduct an ablation study on removing CoT; however, the details on how CoT is removed are not provided. Additionally, the authors could compare their method with an alternative approach that formats all prompts for the four tasks as one input for the LLM to generate the answers simultaneously.

**Questions:**

See in Review.

**Ethics Review Description:**

No ethical issues

**Reviewer Confidence:**

3: The reviewer is confident but not certain that the evaluation is correct

**Scope:**

4: The work is relevant to the Web and to the track, and is of broad interest to the community

---

### Official Review · Reviewer_BAGW · 2023-11-24

**Novelty:** 6
**Technical Quality:** 6

**Review:**

This paper "MMLSCU: A Dataset for Multimodal Multi-domain Live Streaming Comment Understanding" presents a study on live streaming comment understanding using a multimodal approach combining different data streams (audio, transcript, video) etc. for a range of content analysis tasks. The end goal is to provide feedback to the streamers with the above analysis. This is a highly relevant work given the recent popularity of streaming platforms and their massive audience sizes.

The proposed work is interesting from an application standpoint and provides a new and rich multimodal dataset for the research community. I expect that the authors will opensource the underlying dataset to unlock more interesting research directions involving multimodal data, which is a significant contribution to the webconf research community. Thus, I believe it is in the interest of the community to publish this work and enable the authors to distribute their dataset.

I thank the authors for their responses to my suggestions. I believe this work is a valuable contribution to the community, especially since the open sourced dataset and research direction will spur further work. This is an important area in the future since streaming platforms are increasingly mainstream sources of entertainment for younger audiences, and will require AI safeguards and AI moderators at scale. Comment analysis is a valuable input to any such analysis or real-time systems powering streaming platforms. I believe this paper should be accepted.

**Questions:**

1) Could you discuss any specific insights or surprising findings that emerged from adopting a multimodal approach as opposed to a unimodal approach in understanding live streaming comments?

While the proposed method clearly outperforms unimodal studies, a few examples where the multimodal analysis changes a conclusion would be interesting.

2) Are there plans to expand the dataset or the methodology to include additional modalities (e.g., emoticons, user interaction data) or to explore deeper contextual understanding (e.g., sarcasm detection, cultural nuances)?

3) What were the most significant challenges you faced during this research, particularly in terms of data collection and model training? Are there any limitations of your current approach that future research might aim to address?

**Ethics Review Description:**

Ethics review not required since the comment data is anonymized.

**Reviewer Confidence:**

3: The reviewer is confident but not certain that the evaluation is correct

**Scope:**

3: The work is somewhat relevant to the Web and to the track, and is of narrow interest to a sub-community

---

### Decision · Program_Chairs · 2024-01-22

**Decision:**

Accept (Oral)

**Comment:**

The authors introduce a new benchmark dataset for multi-modal learning as well as a method using large pretrained models and chain-of-thought prompting. Model enhances live streaming comment understanding, outperforming text-only approaches. The work is of high quality, it is clear an there is originality.

 + The paper includes a new and rich multimodal dataset that may advance the research in the field.
 + A novel application of CoT prompting.

 - Data annotation process should be further clarified in the final version.
 - Quantify the noise in the annotation process as it may be subjective.

 The reviewers are aligned that this work is worth publishing.